# Semi-supervised organ segmentation with Mask Propagation Refinement and Uncertainty Estimation for Data Generation

Minh-Khoi Pham[1,2][0000−0003−3211−9076],

Thang-Long Nguyen-Ho[1,2][0000−0003−1953−7679] ⋆,

Thao Thi Phuong Dao[1,2,3,5][0000−0002−0109−1114],

Tan-Cong Nguyen[1,2,4][0000−0001−8834−8092], and

Minh-Triet Tran[1,2,3][0000−0003−3046−3041]

[1] University of Science, Ho Chi Minh City, Vietnam
[2] Viet Nam National University, Ho Chi Minh City, Vietnam
[3] John von Neumann Institute, Ho Chi Minh City, Vietnam
[4] University of Social Sciences and Humanities, Ho Chi Minh City, Vietnam
[5] Department of Otolaryngology, Thong Nhat Hospital, Tan Binh District, Ho Chi Minh City, Vietnam

**Abstract.** We present a novel two-staged method that employs various 2D-based techniques to deal with the 3D segmentation task. In most of the previous challenges, it is unlikely for 2D CNNs to be comparable with other 3D CNNs since 2D models can hardly capture temporal information. In light of that, we propose using the recent state-of-the-art technique in video object segmentation, combining it with other semi-supervised training techniques to leverage the extensive unlabeled data. Moreover, we introduce a way to generate pseudo-labeled data that is both plausible and consistent for further retraining by using uncertainty estimation. Our code is publicly available at Github.

**Keywords:** 2D semi-supervised segmentation · Mask Propagation · Uncertainty Estimation

## 1 Introduction

Subclinical examination plays an important role in all medical treatment processes. With the help of deep learning algorithms, human abdominal organs can be identified automatically with effectiveness and efficiency; thus enabling doctors for faster diagnoses. For deep learning agents to achieve high performance, it often comes with a vast amount of high-quality labeled data for the training

---

⋆ The first two authors share the equal contribution.

stage. However, obtaining a sufficient amount of medical data is quite expensive and time-consuming, not to mention the need for medical labels to be evaluated by experts to ensure accuracy for usability. Because of the lack of useful data and scarce medical experts, it makes the problem becomes more challenging to tackle for today's machines.

Since last year, the FLARE22 challenge has introduced a problem in a specific scenario where a shortage of labeled medical data occurs. The included dataset contains only 50 labeled CT volumes whereas 2000 unlabeled others are given. With the provision of an enormous quantity of non-annotated data, participants are required to utilize them to boost the accuracy of their methods for the segmentation task, as well as optimize their solution for practical applicability.

Past solutions mostly approached the problem by inheriting 3D techniques, which usually demand great computing resources. In fact, the original CT volumes must be resampled to a smaller size to fit these 3D-based approaches, then the prediction of these models must also be post-processed back to its preceding sizes, which can damage the precision of the prediction. In terms of that, other teams proposed 2D-based solutions which can leverage the ability to split the CT volumes into batches of slices for efficient processing. But in reality, these techniques face serious performance issues due to the incapability of capturing the temporal information of CT slices. Therefore, to overcome these drawbacks, we propose a novel pipeline, which works completely with only 2D image slices, that can comprehend information from all three planes of a volume.

Furthermore, to make use of a huge number of unlabeled data, two of the most common semi-supervised learning methods are consistency regularization and pseudo-labeling. Consistency-based methods train the model to produce the same pseudo-label for two different views (strong and weak augmentations) of an unlabeled sample, while pseudo-labeling converts model predictions on unlabeled samples into soft or hard labels as optimization targets. However, both of the methods suffer from the noise caused by the model trained on different data distribution (between labeled and unlabeled data). To address the above challenges, we propose a simple technique via modeling uncertainty that can be applied to filter out only potentially good pseudo labels for retraining.

Overall, our main contributions are as follows:

- We propose a 2D-based segmentation pipeline that can fully exploit information of all three dimensions of a CT volume by integrating temporal positional encoding and mask propagation .
- Simple enough, we come up with an uncertainty estimation technique to selectively choose which pseudo labels are useful for next cycle of training.

## 2   Method

### 2.1   Preprocessing

For preprocessing, we apply the Windowing technique [2] with different levels and widths to target specific parts of human organs. Windowing, also known

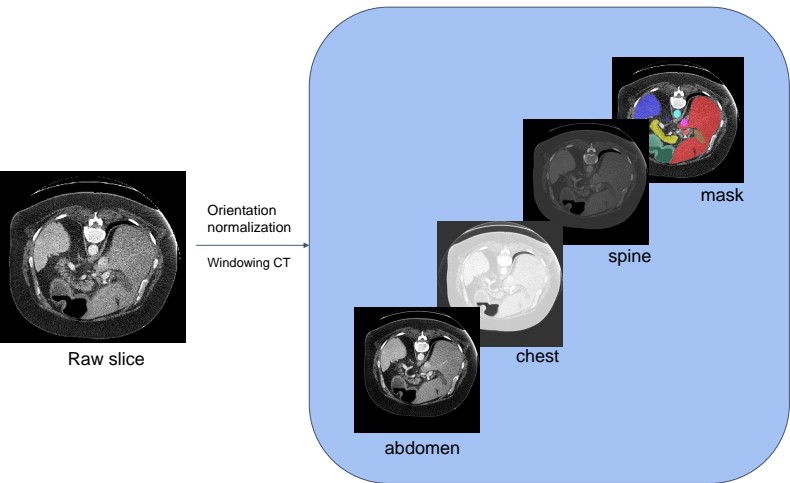

**Fig. 1.** Windowing CT

as grey-level mapping, contrast stretching, histogram modification, or contrast enhancement is the process in which the CT image grayscale component of an image is manipulated via the CT numbers; doing this will change the appearance of the picture to highlight particular structures. The brightness of the image is adjusted via the window level. The contrast is adjusted via the window width. In our experiments, we create 3 different versions of a single slice by highlighting the abdomen, chest, and spine groups and stacking them to one as a three-channel image (Fig. 1).

In addition, we choose the axial plane to cut the slices from the CT volumes since this plane has various dimension sizes. Due to some relatively small organs, it might be better to keep the original size of the slices without any cropping, resampling, or resizing methods. The image is rotated to a predefined angle, then divided by 255 for normalization before going through the next step.

### 2.2   Proposed Method

Our method composes of two main modules: the Reference module and the Propagation module, as can be seen in Fig 2.

In the beginning, we uniformly select only $k$ slices from the CT Volume to be our initial candidates. Next, these slices are processed by using the Windowing technique (described in Section 2.1). Afterward, these slices are put through the Reference module (described in Section 2.2), which performs the standard multiclass segmentation, then the preliminary $k$ masks can be obtained. With these pairs of potential slices and masks as prior knowledge, the Propagation module (described in 2.2) can utilize them to propagate the objects' transformation in-

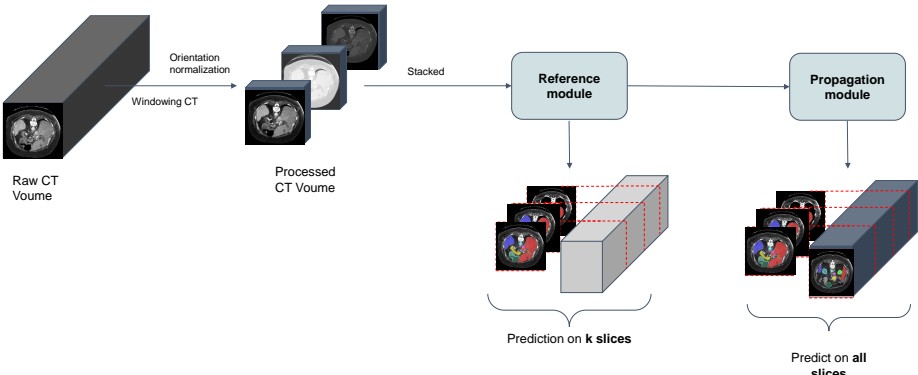

**Fig. 2.** Our overall proposed pipeline. Firstly, the entire CT Volume is processed using windowing CT to get a stack of three-channel slices. Then the slices progress through the Reference module to obtain a minimal number of preliminary masks. Lastly, the Propagation module refine these initial masks to finalize the result.

formation to the remaining slices across the CT volume length. The final output of this module is a 3D dense mask prediction, with each voxel indicating a class.

**Reference module** This module is expected to provide a suggestion of a minimal amount of slices and predicted masks that might contain the most information describing the entire CT Volume. Fig 3 describes the details of this module.

To utilize the enormous number of unlabeled data, we apply the recent semi-supervised method that performs effectively on several other datasets, which is called Cross Pseudo Supervision (CPS) [7] (yellow cube in Fig. 3). CPS enables the usage of unlabeled data by following the dual students technique, where two models are trained simultaneously on labeled data while generating pseudo data for their "peer" to learn. In the testing phase, two models predict the same image, and the result is aggregated by summing up.

We adopt two prominent state-of-the-arts 2D segmentation models with highly different learning paradigms for this CPS framework, which is TransUNet [5] and DeeplabV3+ [6]. While DeeplabV3+ traditionally focuses more on the local information, transformers model the long-range relation, so the cross training can help to learn a unified segmenter with these two properties at the same time. In short, we choose TransUNet and DeeplabV3+ due to their ability to compensate each other for better performance. [13]

In addition, we also propose a both logical and specialist-based strategy to choose which slices can be further used to boost the performance of the Prop-

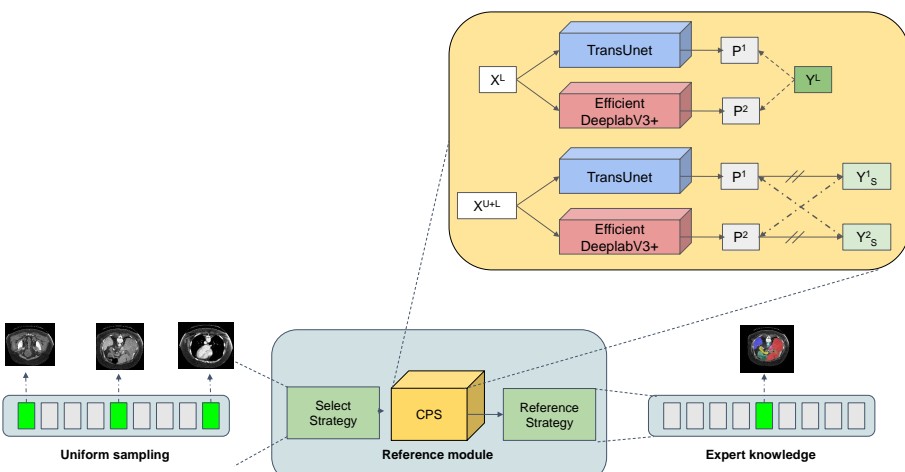

**Fig. 3.** The reference module. The semi-supervised technique CPS is applied in both the training and inference stage to enhance the precision of model prediction. Strategies are used to smartly choose slices that are informative for the next stage.

agation module. The goal of this action is to preserve only some of the most useful information for the refinement stage.

To elaborate on these strategies, prior to being put into the CPS module for prediction, a small number of slices are uniformly sampled from the processed CT volume. After CPS produces segmentation masks for these slices, another selection step is performed to pick only some of the masks that contain the organs having the largest areas.

Although we have employed a semi-supervised learning technique for the Reference module, it still lacks information on the axial plane of the CT volume. Therefore, we simply resolve that by embedding the slices's relative position on the axial dimension as a feature vectors and input them to both networks TransUnet and DeeplabV3+ to learn. The rationale behind this is that, with additional temporal knowledge, models are expected to capture the position constraint for each organ's appearance, hence provide better prediction.

*Positional Encoding*. In order for the model to make use of the order of the sequence, we need to inject some information about the positions of the slices. For simplicity, we add an additional embedding layer to embed the relative position of each slice. Specifically, the embedded position index is concatenated with the hidden features before the final segmentation head. The relative position of $k^{th}$ frame of CT volume $i$ with length $T_i$ is calculated as:

$$PE(k) = \frac{k}{T_i} \tag{1}$$

We attach this layer to both DeeplabV3+ and TransUnet. Since they follow the conventional structure of segmentation models, which comprise of encoding and decoding phases, we manage to attach the layer in a similar way for both of them, as can be described in Figure 4

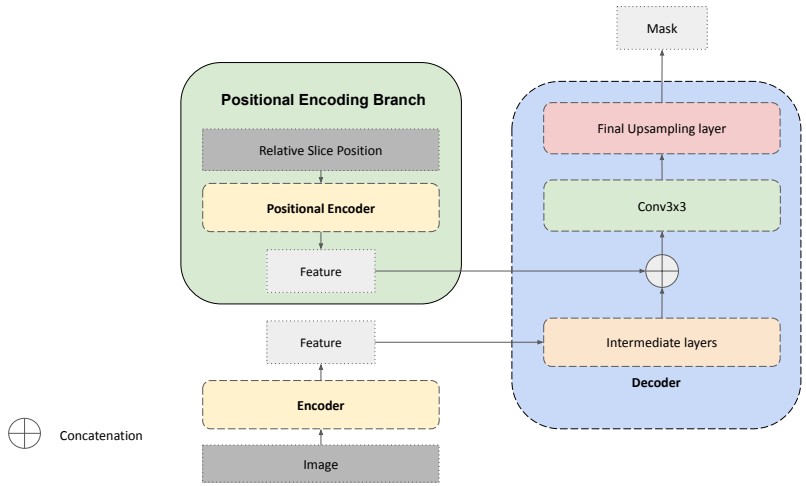

**Fig. 4.** A general and simple way to attach a Positional Encoding branch into segmentation models .

**Propagation module**  This module aims to utilize prior knowledge of given annotated slices from the Reference module to make prediction on the remaining slices, this mechanism can be referred as mask (or label) propagation.

Intuitively, the conventional 2D CNNs cannot comprehend the third dimension information within a CT volume. Thus, in hope of the ability to capture the "temporal" information along the axial plane, we adapt the Space-Time Correspondence Networks (STCN) [8], which is a semi-supervised segmentation algorithm that has achieved promising results on Video object segmentation problem, to this 3D manner.

Basically, STCN proposes the use of a memory bank that stores information about previous frames and their corresponding masks and uses them later as prior knowledge. To generate the mask for the current frame, a pairwise affinity matrix is calculated between the query frame and memory frames based on negative squared Euclidean distance, then it is used for supporting the current mask generation. [8]

Different from the original STCN, we slightly modify it to match the current problem. In the original work, they use only a single dense mask to propagate through the entire video, therefore for the model to perfectly work, that selected

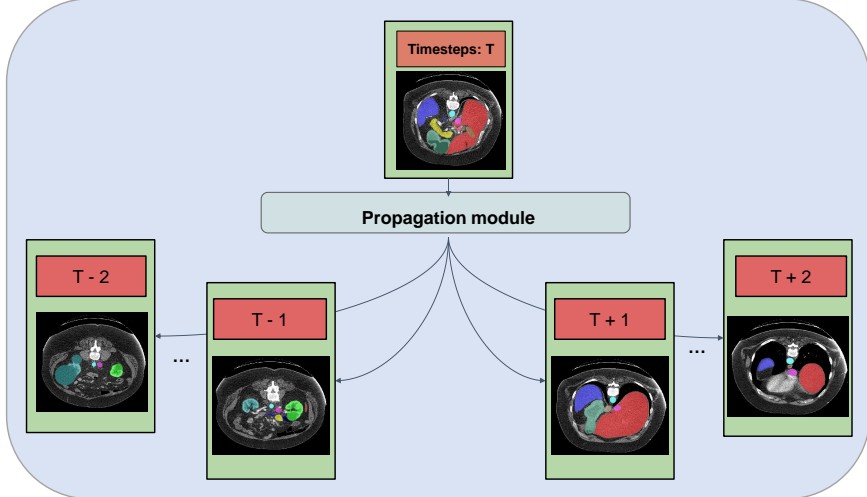

**Fig. 5.** The propagation module. From an annotated slice of CT, at timestep T, STCN can make use of that to spread the information through the entire defined range $[T - k_1, T + k_2]$.

mask must contain information about all available classes. For our case to achieve that goal, we enable the usage of multiple masks for propagation, so that all of these masks should contain enough information about every organs. We also allow the STCN to work in a bidirectional way to enhance the refinement. Fig 5 illustrates this process.

Specially, STCN can be simply trained in the binary manner, meaning that each of the abdominal organs can be learned separately. Therefore, the knowledge can be transferred well between different organ classes.

**Pseudo labeling with Uncertainty Estimation** Given a vast amount of unlabeled CT volumes, we apply a uncertainty estimation technique to effectively maximize the utilization of the data.

Firstly, several CPS models are trained on the provided labeled data. Then, we use these trained CPS models to obtain pseudo masks on the unlabeled set. Inspired from [20], we calculate the dice scores between these pseudo masks and the aggregated one. The mean of these dice scores will be compared with a threshold to determine whether the aggregated pseudo masks are qualified. Simply speaking, consensus-based assessment is used to evaluate the quality of pseudo labels.

We determine a single score for the $i^{th}$ volume in the unlabeled set as the formulation below:

$$score_i = \frac{1}{K \times M} \sum_{k=1}^{K^i} \sum_{m=1}^{M} \mathrm{DSC}(\mathcal{Y}_m^{k,i}, \mathcal{Y}_{AVG}^{k,i}) \qquad (2)$$

$$\mathrm{dsc} = \frac{2|X \cap Y|}{|X| + |Y|} \qquad (3)$$

DSC represents the Dice Score evaluation metric calculating the overlapping area of prediction $X$ and ground truth $Y$. Here $\mathcal{Y}_m^{k,i}$ indicates the $m^{th}$ model's output of the $k^{th}$ slice of volume $i$ while $\mathcal{Y}_{AVG}^{k,i}$ is the mask averaged from all $M$ models for the same slice. The easier the sample is, the more inclined the segmentation models are to get similar outputs. In contrast, hard samples are more likely to be segmented differently by different models. Hence, we use the proposed score to measure the certainty between models' predictions. A higher score gives more credibility to the prediction, as it is more consistent.

All aggregated samples that have high certainty are then reused for the next supervised training cycle. And after the training finishes, the same labeling process is repeated until all aforementioned models achieve satisfactory performance or every unlabeled data has been used.

**Loss function** For the Reference module, we use the prevalent combination of dice loss and cross-entropy loss with smoothing value to alleviate the imbalanced number of the small organs, which occurs due to our splitting into slices process. The same settings are used for CPS in its supervised branch whereas only the dice loss is set up for the unsupervised branch.

For the Propagation module, we implement the online hard example cross entropy (OhemCE or Bootstrapping CE) [21] and also calculate the Lovasz loss [3] at the same time. OhemCE can help reduce the contribution of the background label to the final loss. And since STCN is trained on the binary task, OhemCE can direct the model to focus on visible difficult objects. Meanwhile, Lovasz loss is commonly used in past research and competitions.

### 2.3   Post-processing

We do not use any post-processing techniques because no complex pre-processing ones are used, and we conduct all our experiments on the original-sized image volumes apart from the orientation settings. Thus, before submitting it to the evaluation system, the mask must be transformed back to the original orientation.

### 2.4   Inference Optimization

Unfortunately, we do not apply any engineering technique to reduce resource consumption nor speed up inference process.

## 3   Experiments

### 3.1   Dataset and evaluation measures

The FLARE2022 dataset is curated from more than 20 medical groups under the license permission, including MSD [19], KiTS [10,11], AbdomenCT-1K [15], and TCIA [9]. The training set includes 50 labelled CT scans with pancreas disease and 2000 unlabelled CT scans with liver, kidney, spleen, or pancreas diseases. The validation set includes 50 CT scans with liver, kidney, spleen, or pancreas diseases. The testing set includes 200 CT scans where 100 cases has liver, kidney, spleen, or pancreas diseases and the other 100 cases has uterine corpus endometrial, urothelial bladder, stomach, sarcomas, or ovarian diseases. All the CT scans only have image information and the center information is not available.

The evaluation measures consist of two accuracy measures: Dice Similarity Coefficient (DSC) and Normalized Surface Dice (NSD), and three running efficiency measures: running time, area under GPU memory-time curve, and area under CPU utilization-time curve. All measures will be used to compute the ranking. Moreover, the GPU memory consumption has a 2 GB tolerance.

### 3.2   Implementation details

**Environment settings**  The development environments and requirements are presented in Table 1.

**Table 1.** Development environments and requirements.

| Windows/Ubuntu version | Ubuntu 18.04.5 LTS |
|---|---|
| CPU | Intel(R) Xeon(R) Silver 4210R CPU @ 2.40GHz |
| RAM | 1×32GB; |
| GPU (number and type) | One Quadro RTX 5000 16G |
| CUDA version | 11.6 |
| Programming language | Python 3.10 |
| Deep learning framework | Pytorch (Torch 1.11.0, torchvision 0.12.0) |

**Training protocols**  Currently, we find that using only simple 2D transform functions such as horizontal/vertical flipping or rotating might be enough for both modules to generalize. In the training stage, the Reference module follow traditional training process, in which two models are concurrently trained. For the Propagation module, we inherit the same process as in [8] which samples 3 neighboring slices at a time.

Table 2 and Table 3 mention the training protocols for Reference module and Propagation module, respectively. In both settings, we use the original-sized images, which is [512, 512] for the training and inference phases.

**Table 2.** Training protocols for Reference module: CPS of TransUnet and Efficientnet DeeplabV3+

| Network initialization | Random initialization |
|---|---|
| Batch size | 2 (labeled) + 2 (unlabeled) |
| Patch size | $512 \times 512$ |
| Total iterations | 50000 |
| Optimizer | AdamW |
| Initial learning rate (lr) | 0.0001 |
| Lr decay schedule | multiplied by 0.5 for every iteration at $[40000, 45000]$ |
| Training time | 48 hours |
| Loss functions | Dice Loss + Cross-Entropy Loss |
| Number of model parameters | 105M (TransUnet Resnet50) [6] + 11M (Efficientnet DeeplabV3+) |
| Number of flops | 108G (TransUnet Resnet50) [7] + 1,3G (Efficientnet DeeplabV3+) |

**Table 3.** Training protocols for Propagation module: STCN with Resnet backbone

| Network initialization | Random initialization |
|---|---|
| Batch size | 8 |
| Patch size | $512 \times 512$ |
| Total iterations | 50000 |
| Optimizer | AdamW |
| Initial learning rate (lr) | 0.0001 |
| Lr decay schedule | multiplied by 0.5 for every iteration at $[40000, 45000]$ |
| Training time | 48 hours |
| Loss functions | OhemCE Loss + Lovasz Loss |
| Number of model parameters | 54,416,065 [8] |

## 4  Results & Discussion

**Quantitative results** Here we present both quantitative and qualitative results of our proposed method. We also include the ablation study (Table 5) to further analyze the effectiveness of each of our modules.

Some interesting insights can be spotted in Table 4. Overall, we can see that using the pseudo-labeled data for training, helps boost the performance of the model by a great amount. Unfortunately, we have yet to fully explore every unlabeled sample (only 700 samples were used for training in our submission), but intuitively, the number of used unlabeled samples is likely to be directly proportional to the evaluation result. Another notable observation is that the DSC for some small human organs (gallbladder and adrenal glands) can hardly be improved because of the class imbalance problem (as referred in 4).

**Table 4.** Comparison between using and not using the pseudo labels as supervised training data. The model that is used for the report is TransUnet on public test set. The highlighted figures emphasize the highest values in each row.

| Number of pseudo-labeled samples | 0 | 200 | 700 |
|---|---|---|---|
| Liver | 0.9215 | 0.9555 | **0.9604** |
| Right Kidney (RK) | 0.6548 | 0.7944 | **0.8014** |
| Spleen | 0.8159 | 0.9144 | **0.9255** |
| Pancreas | 0.6235 | 0.7309 | **0.7567** |
| Aorta | 0.8794 | 0.9272 | **0.9335** |
| Inferior Vena Cava (IVC) | 0.7145 | 0.7967 | **0.8207** |
| Right Adrenal Gland (RAG) | 0.4688 | 0.6507 | **0.6545** |
| Left Adrenal Gland (LAG) | 0.4209 | **0.6179** | 0.6138 |
| Gallbladder | 0.4798 | **0.5889** | 0.5885 |
| Esophagus | 0.7086 | **0.7784** | 0.7783 |
| Stomach | 0.7446 | 0.8403 | **0.8424** |
| Duodenum | 0.4387 | 0.5617 | **0.5679** |
| Left Kidney (LK) | 0.6763 | **0.8112** | 0.8026 |
| Mean DSC | 0.6575 | 0.7668 | **0.7728** |

**Table 5.** Ablation experiment on each proposed modules and techniques.

| No. | Positional Encoding | CPS | Uncertainty Estimation | Mask Propagation | Mean DSC |
|---|---|---|---|---|---|
| 1 | | | | | 0.6419 |
| 2 | ✓ | | | | 0.6575 |
| 3 | ✓ | ✓ | | | 0.762 |
| 4 | ✓ | ✓ | ✓ | | 0.7728 |
| 5 | ✓ | ✓ | ✓ | ✓ | **0.784** |

Table 5 shows that each module contributes to the final score of our submission. The baseline model that is reported in the first row is TransUnet. The Cross Pseudo Supervision (CPS) refers to using both DeeplabV3+ and TransUnet as training models. The third and fourth rows where both CPS and Uncertainty Estimation (UE) is used mean that pseudo-labels that are qualified by UE are used as supervised inputs in CPS workflow, whereas the remaining unlabeled data are used as unsupervised inputs. Noticeably, in the fourth row, with the Mask propagation (MP) applied, DSC score is enhanced substantially. It is surprising that MP only looks upon the minority of the slices to fully propagate through the whole volume. The detailed evaluation for our best submission is shown in Table 6.

**Qualitative results** Looking at examples that are well-predicted by our approach in Fig 6 (1b, 2b, 3b), it demonstrates good segmentation masks with clear and smooth mask boundaries. Some small organs can also be seen segmented successfully and precisely meaning that both proposed modules can work effectively with organs having various sizes.

**Table 6.** The final evaluation score for our final submission.

| Classes/Metrics | DSC | NSD |
|---|---|---|
| Liver | 0.974 ± 0.036 | 0.963 ± 0.063 |
| Right Kidney (RK) | 0.883 ± 0.233 | 0.868 ± 0.241 |
| Spleen | 0.9494 ± 0.115 | 0.935 ± 0.134 |
| Pancreas | 0.772 ± 0.147 | 0.877 ± 0.145 |
| Aorta | 0.96 ± 0.045 | 0.976 ± 0.06 |
| Inferior Vena Cava (IVC) | 0.86 ± 0.123 | 0.86 ± 0.143 |
| Right Adrenal Gland (RAG) | 0.735 ± 0.138 | 0.855 ± 0.144 |
| Left Adrenal Gland (LAG) | 0.69 ± 0.171 | 0.816 ± 0.2 |
| Gallbladder | 0.75 ± 0.313 | 0.733 ± 0.328 |
| Esophagus | 0.783 ± 0.147 | 0.88 ± 0.143 |
| Stomach | 0.86 ± 0.113 | 0.84 ± 0.142 |
| Duodenum | 0.6 ± 0.2 | 0.79 ± 0.215 |
| Left Kidney (LK) | 0.877 ± 0.22 | 0.863 ± 0.23 |
| Mean | 0.8233 | 0.8668 |

On the other hand, our models suffer from various difficult cases where organs are missing. Generally, there are two cases that negatively affects our approach:

1. Relatively small organs (adrenal glands (Fig 6 (1e)), gallbladder (Fig 6 (1e)), and esophagus (Fig 6 (3e))) account for the lowest DSC since they usually are failed to be identified by the Reference module.
2. Other organs (pancreas (Fig 6 (1e)) and duodenum (Fig 6 (2e))) despite having larger size, yet their lengths on the axial plane are short and sometimes occluded by many surrounding organs, which can affects how the information propagating through the slices, causing class confusion in the result.

Furthermore, due to the our two-staged pipeline, for the results of the second stage to be good really relies on the first stage' performance. If the reference stage miss-segments any organ, that one will be missed during the entire propagation process. Having said that, this issue mostly just occurs to organs that have short-size length on the axial plane.

**Efficiency results** Segmentation efficiency results are reported in Table 7. GPU memory and GPU utilization is recorded every 0.1s. The Area under GPU memory-time curve and Area under CPU utilization-time curve are the cumulative values along running time.

**Table 7.** Efficiency evaluation from official report.

| Running times (s) | AUC GPU | AUC CPU |
|---|---|---|
| 140.73 | 647605 | 3729 |

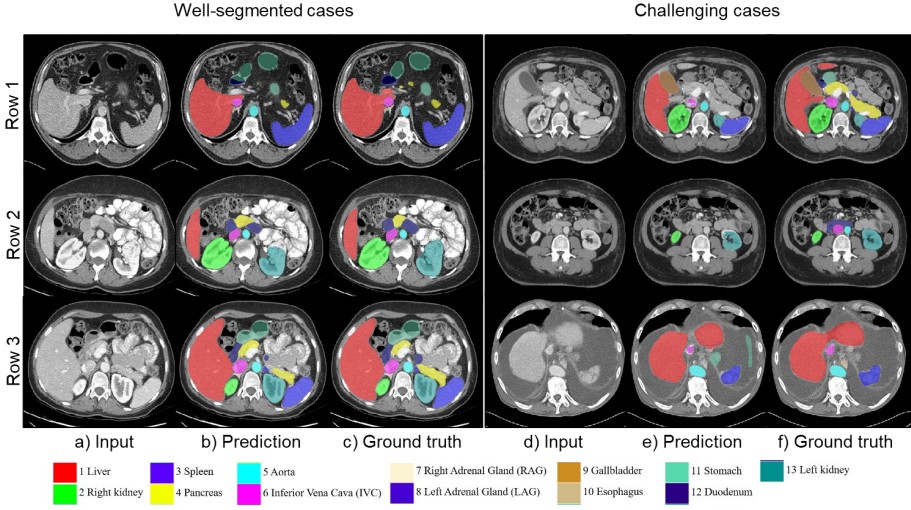

**Fig. 6.** Qualitative results from the validation set. We illustrate both well-segmented and challenging examples for our proposed segmentation pipeline

**Limitation and future work** Apparently, although our proposed method has yet to achieve the high result, we believe it can be further improved if these limitation that we identify here are solved. First of all, the problem of imbalanced dataset has arisen because we perceive this as a 2D problem. Due to the slices splitting process, small organs (such as pancreas, gallbladder or adrenal glands) only appear in a small amount of slices, while larger objects have wider range of appearance. Therefore, it leads to the problem of imbalanced dataset. We tried some ways to tackle the problem, for instance: smart sampling, or imbalanced loss, however only slightly improvement was seen. Secondly, the proposed approach is a two-stage method, the second stage is undoubtedly dependent of the first one. If there are any organs that are missed by the Reference module, it definitely cannot be recovered in the Propagation phase. Thus, more attention is needed for the Reference module. In the future, it is encouraged to focus on boosting the performance of the Reference module by fully exploiting the temporal information.

## 5 Conclusion

In summary, with new advancements in technology, there are endless possibilities for what can be achieved. In the medical field, one of the most common problems that doctors face is accurately segmenting 3D objects from 2D images or volume sequences. Recently, we propose a novel two-stage pipeline that can leverage the strength of many state-of-the-art 2D deep learning algorithms and techniques in videos and images, into the task of 3D object segmentation. This proposal aims to

introduce a novel and inspirational approach to solving one of the most common problems in the medical field. In addition, to break the barrier of differences in medical pipeline processes, our solution is able to transfer and exploit the power of multiple domain data in datasets creating more accurate results.

## 6   Acknowledgements

This research is funded by Vietnam National University Ho Chi Minh City (VNU-HCM) under grant number DS2020-42-01

The authors of this paper declare that the segmentation method they implemented for participation in the FLARE 2022 challenge has not used any pre-trained models nor additional datasets other than those provided by the organizers.

Furthermore, no manual intervention has been made in the contribution to the results of the proposed method

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
