# OpenReview forum: "Semi-supervised organ segmentation with Mask Propagation Refinement and Uncertainty Estimation for Data Generation"
_MICCAI.org/2022/Challenge/FLARE_

### Official Review · Reviewer_8ZwF · 2022-09-17
**Well-written paper with sufficient ablation experiments**

**Rating:** 8
**Confidence:** 4

**Review:**

**Summary:**

This work applies Windowing CT for data augmentation, employs DeeplabV3+ and TransU-Net for 2D segmentation task, and adopts CPS for semi-supervised training. A two-stage pipeline is proposed, including the reference module and propagation module, which is clearly explained in section 2. This work is a well-written paper, and its experiments are sufficient and substantial.

**Suggest improvements:**

As discussed in the 'Reference module' section:

> Embedding the slices’s relative position on the axial dimension as a feature vectors and input them to both networks TransUnet and DeeplabV3+ to learn.

How are slice position embedding features fused with feature maps, and how to train the embedding features? It is not explained clearly, and there is no ablation experiment to prove its effectiveness.

---

> ### Author Response · Authors · 2022-10-14
> **Revision has been updated**
>
> Thanks for your comment.  Detailed about "Slice position embedding features" has been updated and described in a figure. Ablation study is also provided.

---

### Official Review · Reviewer_6fQc · 2022-09-20
**Basically complete but with some deficiencies**

**Rating:** 6
**Confidence:** 3

**Review:**

In this paper, the authors adopted mask propagation refinement and uncertainty estimation for CT volume labeling to perform semi-supervised abdominal organ segmentation.

The manuscript is basically complete. Suggestions or deficiencies:

1. In Table 2 and Table 3, the authors should describe the patch size, which seems to be 512×512.
2. In Table 2 and Table 3, the authors should give a more detailed description of Random initialization.
3. In Figure 5, the authors should point out which image each row comes from.
4. Tables 2 and Tables 5 are too wide.
5. The authors should divide section 4 into different subsections.
6. The authors should add segmentation efficiency results and segmentation efficiency analysis in section 4.

---

> ### Author Response · Authors · 2022-10-14
> **Revisions has been updated**
>
> Figures and Tables are updated accordingly. Efficiency analysis will be updated shortly. Thanks for your detailed comments

---

### Official Review · Reviewer_qCNZ · 2022-09-20
**Complex 2D model, results are not bad, but are not SOTA**

**Rating:** 6
**Confidence:** 3

**Review:**

Pros:
- the results show that the proposed semi-supervision really matters
- authors put an effort to implement such a complex method

Cons:
- results are not SOTA
- no comparison with 3D-conv models

---

> ### Author Response · Authors · 2022-10-20
> **Response of Authors**
>
> Thanks for your review. It is unfortunate that the results are not SOTA but is predictable. Therefore, we just want to demonstrate that 2D methods are also a promising field to be explored in the future, since researchers has been exploiting the same "no-new" architecture and idea ceaselessly in the problem of 3D medical segmentation. Hopefully, this will open for many new improvement in the future.
> And due to the limit of time, our team cannot conduct experiment regarding the 3D-conv models.

---

### Official Review · Reviewer_hMfG · 2022-09-21
**Try on well-known 2D segmentation method and video processing method on a medical 3D image segmentation**

**Rating:** 6
**Confidence:** 5

**Review:**

A 2D segmentation pipeline was proposed in the article, with a reference module and a propagation module. A method of how to choose pseudo labels was tested.

Overall, using 2D segmentation model plus video processing technique is not very suitable for medical image processing.
Many models have been tested on medical image segmentation, While 3D segmentation models had much better performance, comparing with general 2D segmentation model.  Low baseline value in Table 5 with DeeplabV3+ model, was also proved this point.

Advise to start with some models, commonly used in medical image analysis.

---

> ### Author Response · Authors · 2022-10-20
> **Response from Authors**
>
> Thanks for your review and suggestion. We actually looked for models in medical field and found TransUnet, which has been used in medical area in the original work. But, we agree that more common models in medical field should be explored for better analysis.....

---

### Meta-Review · Program_Chairs · 2022-09-28

**Recommendation:** Major Revision
**Confidence:** 5

**Metareview:**

Please adjust the window width and level of CT images to 40 and 400, respectively in all figures.
Reviewers raise many concerns and suggestions. Please address all comments in the revised manuscript.

---

> ### Author Response · Authors · 2022-10-14
> **Revision has been updated**
>
> All figures are updated accordingly. All concerns are resolved